# Soda lignin as a sustainable photosensitive component for conventional and controlled radical photopolymerization

Min Wang[1,4], Xiongfei Luo[1,2,4], Qunying Wang[3], Veronika Strehmel[3], Zhijun Chen [1] ✉, Shujun Li [1] ✉ & Bernd Strehmel [3] ✉

Valorization of waste for chemical processes addresses a challenging task for the society. Soda lignin (**AL**) available in the pulp industry operated in a photocatalytic cycle $CuX_2/L$ (X: Br, Cl; **L**: $Me_6$TREN: Tris[2-(dimethylamino)ethyl]amine) and to initiate radical photopolymerization of methyl methacrylate (MMA) with α-bromophenylacetate (EBPA) by exposure at 420 nm. This resulted in poly(methyl methacrylate) (PMMA**)** exhibiting a dispersity <1.3 with $CuX_2/L$ loading of ≲52 ppm. Deactivator reduction increased dispersity. MMA, styrene, and benzyl methacrylate successfully served as monomers for chain extension and block copolymerization experiments. This worked with $CuX_2/L ≥ 6.5$ ppm, although conventional radical polymerization additionally competed. Adding of sodium pyruvate enabled radical photopolymerization under air. Modification of lignin with an aryl sulfonate group resulted in a one-component photoinitiator. It successfully initiated radical photopolymerization of a monomer mixture of (hydroxyethyl)methacrylate (HEMA) and urethane dimethacrylate (UDMA). Scale-up experiments approved practicability of photo-ATRP.

After the first demand to use lignin as an important renewable resource in 1973[1,2], it has taken several decades to recognize its value. This biomaterial addresses the necessity of more research for integration into value chains[2,3]. Lignin has received increasing interest in the refinery of biomaterials[2–10]. Its abundance in wood provides a valuable chemical source compared to fossil materials[4,9,11–13]. More interesting appears the fact that **AL** is also available as waste in the woodworking industry. The paper manufacturing industry[7–9,12,13] can be seen as one source where huge amounts of **AL** have been generated; that is, depending on the source of 50–70Mt/year[13] or 100 Mt/year[12]. Their use focused on other fields as a source to generate energy by combustion, synthetic gas production, as a binder or cement dispersant[3]. Research also focused on synthesis of materials available by degradation such as aldehydes, phenolic materials, hydrocarbons, urethanes or epoxides[3]. Natural availability and, therefore, low toxicity of – explain the interest in biomedical applications. Other applications include fire retardants, sequestrants, nanomaterials, and energy storage materials[3].

Despite these efforts, the use of waste-related lignin in value chains as light sensitive component can still be seen in its infancy[14–16]. Here, its functionalization possesses potential, as recently shown, where demethylated lignin operated as a photocatalyst combined with $Fe_3O_4$ as a cocatalyst in conventional radical photopolymerization and water cleaning based on a Fenton protocol[17].

Lignosulfonate operating in 3D printing[18,19] represented another alternative for use as photoinitiator. Room temperature delayed fluorescence and phosphorescence[18–22] of the cured material provided additional features to develop materials for fraud protection[18,19,22]. This demonstrates again the potential for use in photopolymerization as green technology[23,24] for 3D printing[18,25–32], chemical drying of coatings[33] or to process electronic materials[34].

Conventional radical photopolymerization ("free radical polymerization": outdated by IUPAC terminology[35]) would benefit from further developments of sustainable light sensitive materials. It has continuously grown in Asia, Europe, and North America in the last 10 years without

[1]Key Laboratory of Bio-based Material Science & Technology, Northeast Forestry University, Ministry of Education, Harbin, China. [2]College of Chemistry, Chemical Engineering and Resource Utilization, Northeast Forestry University, Harbin, China. [3]Department of Chemistry, Institute for Coatings and Surface Chemistry Niederrhein University of Applied Sciences, Adlerstr. 1, Krefeld, Germany. [4]These authors contributed equally: Min Wang, Xiongfei Luo. ✉e-mail: chenzhijun@nefu.edu.cn; lishujun@nefu.edu.cn; bernd.strehmel@hs-niederrhein.de

showing a break in the pandemic[36]. Main challenges address to adapt systems operating with modern LED systems since the use of mercury-based radiators will be prohibited in the EU starting from 2027[37]. Low toxicity would be another challenge to replace photoinitiators, causing toxicological issues[38,39]. Here, heterogeneous systems also possess big potential either as one or multi-component system because of the low cytotoxicity and low migration tendency as shown for carbon nanodots[40–45].

Lignin represents an interesting candidate for making photoinitiating materials following either a photocatalytic mechanism based on light-mediated Atom Transfer Radical Polymerization (photo-ATRP) or conventional radical photopolymerization. The heterogeneous habitus, the high molecular weight, and low migration tendency can contribute to solve toxicological problems of today's traditional photoinitiators.

## Results

### Motivation and analysis of AL

This study focused on different protocols following either conventional radical polymerization or radical photopolymerization with ≲50 ppm CuBr$_2$/**L** according to a photo-ATRP setup[40,46–48] with a 420 nm emitting LED. The increased reports of toxicological issues of traditional radical photoinitiators[38,39] have initialized the development of alternative materials[49] where fewer problems would be expected. Lignin appears as a reasonable substrate since its biological origin classifies it as a material with no toxicological issues. Motivated by this, attention has been paid to the valorization of technical lignin available as waste; that is soda lignin/alkaline lignin (**AL**).

**AL** mostly operated as heterogeneous material dispersed in the reaction mixture, Fig. 1a. The size was between 100-300 nm. Figure 1b illustrates a general pattern of lignin with some respective structures. The complex supramolecular structural features formed by the aromatic moieties can be seen as one source to initiate photonic events; that is for example, long afterglow[18–20,22,50]. Thus, the potential of lignin to sensitize radical photopolymerization can be seen as a logical consequence[16,18,19,51]. It can operate in a photocatalytic cycle following a protocol based on light-mediated Atom Transfer Radical Polymerization (photo-ATRP)[40,46,48], where photoinduced reduction of the deactivator (CuX$_2$/**L**) mainly controls the efficiency of activator (CuX/**L**) formation. Figure 1c exhibits the necessary chemicals for this protocol comprising ethyl α-bromophenylacetate (EBPA), the cocatalyst CuBr$_2$/**L** operating as deactivator complexed with the amine ligand **L** (**L**: Me$_6$TREN, PMDETA: $N$, $N$, $N'$, $N''$, $N''$- Pentamethyl diethylenetriamine) to make it compatible in the respective monomer/solvent mixture. The structural features shown in Fig. 1d additionally facilitated aryl sulfonation, resulting in a one-component photoinitiating system following Norrish I cleavage. The low ability to initiate conventional radical required this modification.

X-ray photoelectron spectroscopy (XPS) survey (Fig. 2a) showed C and O as major elements. Sodium originated from the paper-making process. High-resolution XPS exhibited four peaks in the C1s spectrum (Fig. 2b) appearing at 284.8 eV, 286.3 eV, 288.3 eV and 289.1 eV assigned to C-C, C-O-C, C=O and O-C=O moieties, respectively. Signals at 530.9 eV, 531.7 eV, and 535.4 eV in the O1s spectrum (Fig. 2c) related to C=O (carbonyl), C-OH/C-O (hydroxyl, ether), and O-C=O (ester) moieties, respectively, in AL. The FTIR spectrum (Fig. 2d) showed a broad band indicating the OH vibration (~3359 cm$^{-1}$). The vibration at 1120 cm$^{-1}$ is related to the C-O-C ether structures and characteristic peaks (~1600 and ~1510) of the benzene ring.

In addition, photocurrent experiments (Fig. 2e) did not give any evidence regarding the formation of long-living charges in the dark period[52–54]. Room temperature phosphorescence (RTP), and therefore the formation of triplet states, was not observed either. Thus, the paper manufacturing process can be seen as a one reason to contribute for the loss of supramolecular structures due to the alkaline treatment. Supramolecular structures contribute to long afterglow and thus to the formation of triplet states[18–20,22,50], and charge migration events[52,53]. Absorption spectroscopy demonstrated an increase of absorption upon the addition of CuBr$_2$ (Fig. 2f) indicating binding/interacting with lignin.

Absorption spectroscopy and electrochemical properties answered questions related to the absorption efficiency and capability to donate an electron in photoredox cycles (Figs. S1a, b for more details in Supplementary Information). Cyclic voltammetry (CV) analysis showed an oxidation peak ($E_{ox}$) at 1.15 V in the case of **AL** (4 mg/mL), which gradually shifted to lower values after the addition of Cu(II); that is an $E_{ox}$ of 1.09 V with 25 ppm CuBr$_2$/Me$_6$TREN, 1.05 V with 52 ppm CuBr$_2$/Me$_6$TREN), and 0.99 V with 100 ppm CuBr$_2$/Me$_6$TREN, see Fig. S1c in Supplementary Information. This shows an increase of the oxidation capability of lignin moieties interacting with CuBr$_2$. Switching to CuCl$_2$/Me$_6$TREN connected to an additional decrease of $E_{ox}$ of 0.87 V with 31 ppm loading of CuCl$_2$/Me$_6$TREN. Reduction showed a change of the $E_{red}$ peak to −0.7 V after adding of 31 ppm CuCl$_2$/Me$_6$TREN to **AL** (Figs. S2–3 in Supplementary Information). These data are needed for the calculation of the free reaction enthalpy of the photoinduced electron transfer ($\Delta G_{el}$) *vide infra* according to Eq. 1 ($F$ = Faraday constant, $E_{00}$ = excitation energy).

$$\Delta G_{el} = F \times (E_{ox} - E_{red}) - E_{00} \qquad (1)$$

The $\Delta G_{el}$ calculation resulted in a negative value for the reaction of the photoexcited donor **AL** and CuBr$_2$/Me$_6$TREN operating as an oxidizing agent. AL, depending on the loading with CuBr$_2$/Me$_6$TREN described *vide supra* an oxidation potential with an oxidation potential between 1.15 V to 0.99 V, an $E_{00}$ value of 2.36 eV (525 nm), and a reduction potential of –0.39 V[55] for (Cu/Me$_6$TREN)Br$_2$. The $\Delta G_{el}$ resulted between −0.82 eV to −0.98 eV, respectively. This quantity indicated that photoinduced electron transfer (PET) can proceed from a thermodynamic point of view, where lignin operates as a photosensitizer to initiate the photocatalytic cycle of the ATRP. This does not include considerations of internal activation barriers[56] in the PET protocol. The redox potentials depend on the concentration of CuBr$_2$/L affecting somehow the standard potential. This also affects the ATRP equilibrium constant discussed *vide infra*. A lower copper concentration may drive the scenario towards chain termination and therefore increase of dispersity. Thus, one needs to compromise between the required uniformity of the polymer and the acceptable amount of copper catalyst.

### Feasibility to operate as a one-component photoinitiator

Previous studies of lignin reported about the availability of Aryl-CO-R moieties in lignin (Scheme 1)[57]. This group can homolytic cleave, resulting on Aryl-CO· and R·, which shall initiate conventional radical polymerization. It should enable this biomaterial as a source to generate initiating radicals following Norrish I cleavage resulting on enabling this material as a biobased one-component initiator. However, the results obtained evidenced no effective initiation of conventional radical photopolymerization using a monomer mixture consisting of HEMA and UDMA. Here, the addition of HEMA improved the performance to disperse lignin in UDMA. Nevertheless, **AL** exhibited insufficient initiation efficiency in the HEMA/UDMA mixture. However, these radicals have a lower mobility because they originate from polymer matter and should therefore result in a significantly lower initiation efficiency. The higher molecular weight of these radical connects to lower diffusion constants and there less reactivity. More target-oriented was the reaction between **AL** and benzene sulfonyl chloride, resulting in aryl sulfonic esters (AL–SO$_2$Ph). Here, bond cleavage can successfully form low molecular PhSO$_2$· radicals exhibiting higher mobilty[49]. This could be a feasible approach to compensate the drawbacks *vide supra*. Such a treatment additionally increased the hydrophobicity, which connects to an improved dispersibility of AL–SO$_2$Ph in the crosslinking monomer mixture. Its exposure resulted in conventional radical photopolymerization of HEMA/UDMA, Fig. 3. A decrease in initiator loading did not strongly affect the final version, which appears more typical for traditional photoinitiators[43]. Such systems are much more complicated than low molecular weight photoinitiators because the heterogeneous nature of such heterogeneous systems can lead to composite-like materials after irradiation. This has been shown in a previous study. A forthcoming study will report on the progress in this field.

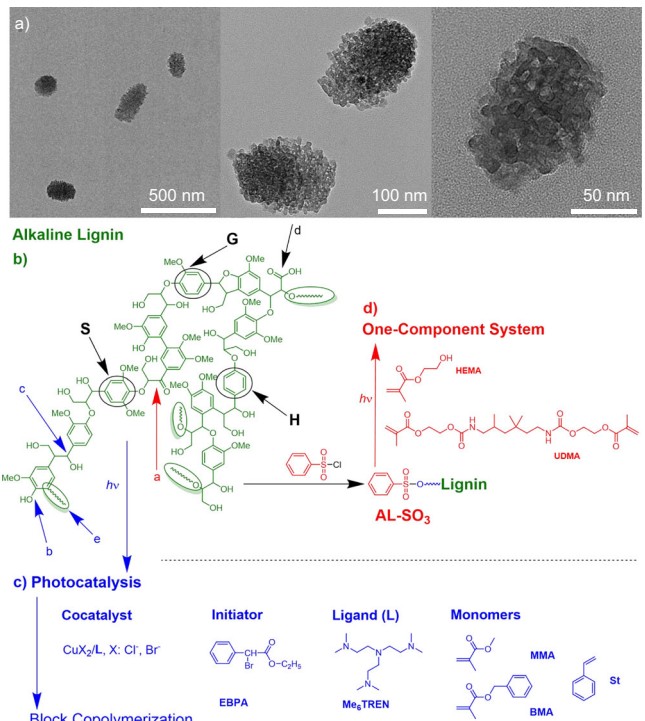

**Fig. 1 | Size and structural sketches for AL including its operational function.**
**a** Transmission electron microscopy (TEM) of **AL** in dimethyl sulfoxide (DMSO) with different magnifications. **b** Sketch for alkaline lignin. **c** Photoexcited **AL** additionally enables radical polymerization based on a photocatalytic mechanism with alkyl bromides as initiator and CuBr$_2$/**L** as cocatalyst. Me$_6$TREN was the ligand **L**. **d** Functionalized **AL** (AL-SO$_3$) initiates conventional radical photopolymerization of urethane dimethacrylate (UDMA) and hydroxyethyl methacrylate (HEMA) as one-component photoinitiator (AL-SO$_3$).

AL without functionalization demonstrated no notable activity under our experimental conditions, despite its photocleavable groups. This is likely lower compared to those photoactive groups obtained after functionalization. Functionalization of the phenolic OH groups with photocleavable moieties is clearly beneficial because it reduces the number of hydroxyl groups in the photoactive material, thereby minimizing interference from conventional radical polymerization. Again, this preliminary result demonstrates the potential of **AL** to serve as natural feedstock to design photoinitiating one-component photo-initiating materials but it keeps in mind that more research will be required to discover the full potential.

## Operation as photosensitizer in photocatalytic cycles for photoinitiation

A photo-ATRP protocol, as shown in Fig. 4a, served as the fundamental to design the photocatalytic system (Eq. 2a–d)[58]. According to the spectroscopic results disclosed in Figs. S4–7 in Supplementary Information, additional interactions shall be included that relate to the adsorption/desorption of CuBr$_2$/L on the surface of **AL** (Eq. 3). This process needs more consideration in future works because it also relates to reactivity and reaction conditions of the ATRP proceeding nearby the particle. Such explorations can be still seen its infancy although some pioneering works already appeared. In addition, the supramolecular structure relating to photoreactivity[19,22] may depend on the lignin batch, which related to waste in this study[40–42,52,53,59]. Thus, the source and method to get it may additional parameter determining microstructure of the surface.

Ion exchange by phenolic groups (Eq. 4) may be one possibility influencing the photolytic rate of activator formation and deactivator back formation (Eq. 2b) followed by chain propagation (Eq. 2c). Binding of reactants on the surface would also impact the reactivity because adsorption effects mobility, and ion exchange the equilibrium constant of ATRP ($K_{ATRP}$). Figure 4b shows a TEM picture of an **AL** particle isolated after photopolymerization. It exhibits a larger size after the reaction, as approved by light scattering measurements, see Fig. 4c. The inclusion of a swollen polymer into the free spaces of the particle may additionally explain this

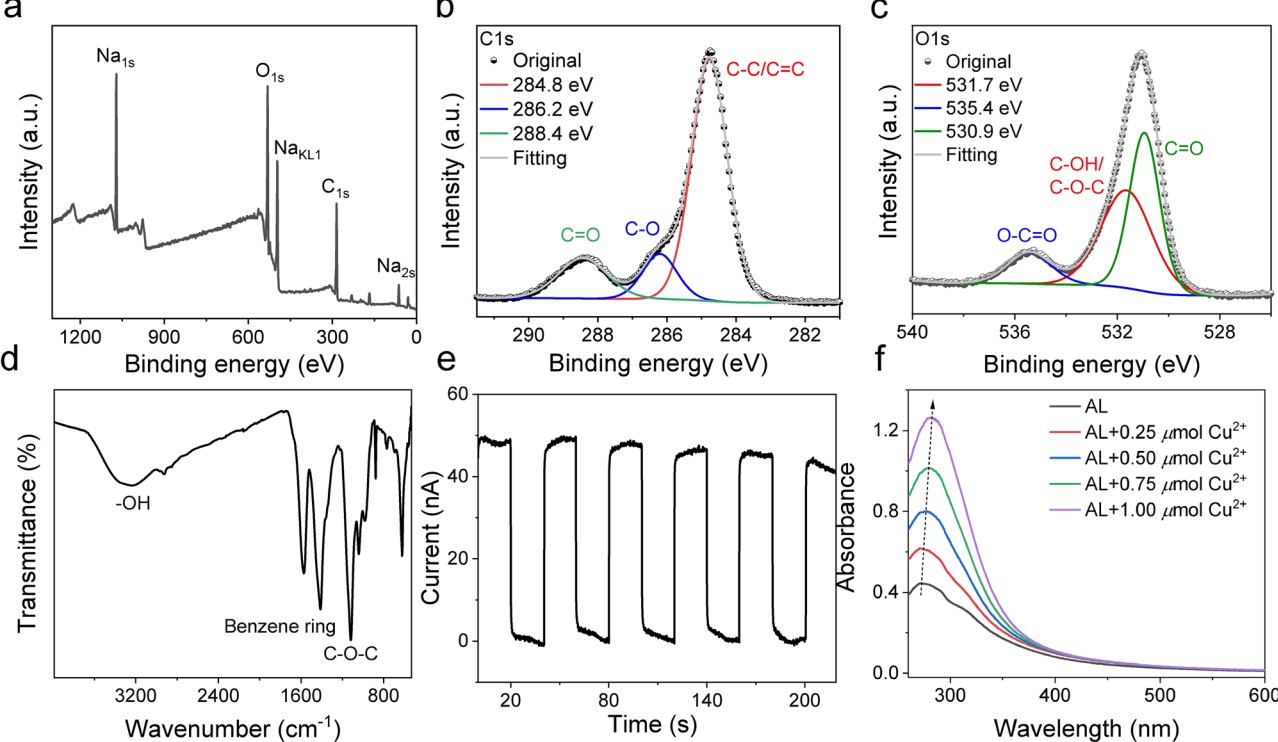

**Fig. 2 | Composition of AL. a** XPS survey spectrum; **b, c** high-resolution XPS spectra of C1s and O1s; **d** FT-IR spectrum of AL; **e** photocurrent of **AL** under xenon lamp; **f** UV-Vis absorbance spectra of **AL** and loading of different content of CuBr$_2$.

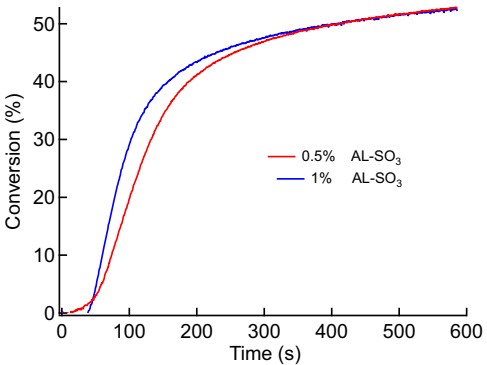

**Fig. 3 | Photoinitiation of monomers by AL-SO₃.** Double bond conversion of aryl sulfonated lignin (AL-SO₃) in radical polymerization using a mixture of HEMA and UDMA (λ = 365 nm, $I$ = 35 mW/cm²).

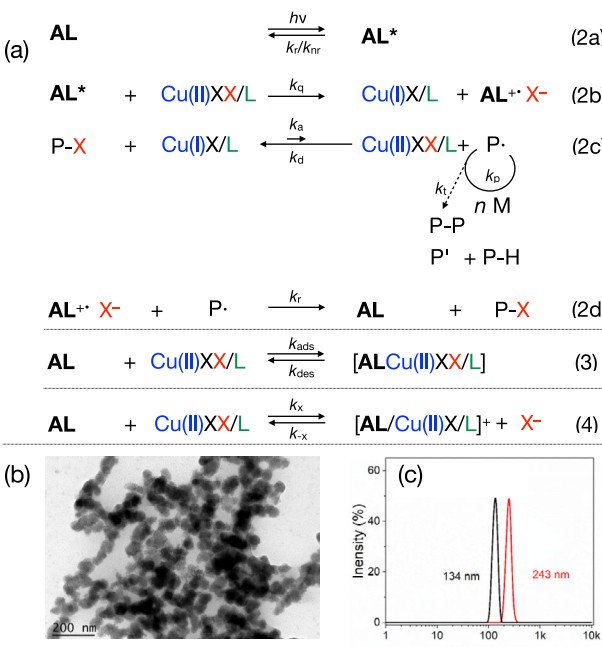

**Fig. 4 | Proposal for different interactions between phenolic moieties of AL and CuBr₂/L using an ATRP cocktail. a** Sensitized light-mediated ATRP with traces of deactivator (CuBr₂/L) in the ppm region ($k_a$ = activation rate constant, $k_d$ = deactivation rate constant, $k_p$ = propagation rate constant), modified from ref. 58. This equation importantly affects the efficiency of photo-ATRP also in heterogeneous systems. Here, additional conditions must be considered that are influenced by different interactions with lignin. This includes: (1) full exchange between Br⁻ and phenolate, (2) partial exchange between Br⁻ and phenolate, (3) no exchange between Br⁻ and phenolate. (4) relates to metal-free photo-ATRP based on an oxidative mechanism. **b** TEM picture of an isolated **AL** particle after photo-ATRP with MMA. **c** particle size of **AL** before (black) and after (red) photo-ATRP with MMA.

phenomenon. This clearly increases viscosity around the particle surface because there is less efficient mixing with the solvent. Different reactivity conditions exist because the reactions proceeding in the cavities of the particle are different from the reactions in the surroundings outside of the particle.

Table 1 summarizes the polymerization results *vide infra*. Entries 1-3 show a decrease in conversion, the number average molecular weight ($M_n$), and initiation efficiency $I_{eff}$[60] with increasing initiator concentration, while the dispersity ($Đ$) only slightly responded. Repeating experiments confirmed reasonable reproducibility. Furthermore, a higher **AL** concentration (Run 4) resulted in higher conversion while the dispersity increased, confirming less uniformity. A lower **AL** loading, as shown in run 5, resulted in less conversion and lower $I_{eff}$. The changed absorption conditions mainly affected these changes compared with run 1. Dispersity increased with decreasing **L** and CuBr₂ to 25 ppm (run 6), while increasing **L** and CuBr₂ to 102 ppm (run 7), $M_n$ remained like run 6 but the conversion was the largest.

A higher concentration on initiator and deactivator, namely CuBr₂/Me₆TREN, in run 8 resulted in lower conversion, $M_n$, and $Đ$. A change of MMA by Styrene (St) led to a lower conversion while the $Đ$ exhibited the best uniformity of molecular weight distribution (run 9). Change of **L** by PMDETA (run 10) approved less uniformity and $I_{eff}$ compared to run 1. The reduction potential of EBPA should enable PET with photoexcited **AL**, resulting in initiating radicals that shall deactivate with less selectivity in an oxidative metal-free photo ATRP scheme[53,61]. Surprisingly, additional experiments without copper catalyst and **L** (run 11 and 12) resulted in polymers accompanied by high conversion and large $Đ$ with no possibility of extending the chain, see Fig. S8 in Supplementary Information. Thus, chain termination efficiently competed and assigned both examples to conventional radical photopolymerization. Here, the alkyl bromide operated as a coinitiator following a classical multi-component photoinitiator setup. Nevertheless, such a system may appear interesting for several uses in practice due to the low toxicity of **AL** and the macroinitiator because both are polymers with low migration potential. In addition, a comparative experiment (run 13) without **AL** showed no polymerization, confirming that the excitation of CuBr₂/Me₆TREN does not have an impact on the results here. Apparently, the addition of (Cu/Me₆TREN)Br₂ suppresses the dominance of chain terminations as documented by the lower dispersity. On the other hand, photopolymerization with **AL** and FeBr₃ resulted in a polymer with much higher $M_n$ and $Đ$, indicating the dominance of chain termination and therefore conventional radical polymerization. Thus, choosing (Cu/Me₆TREN)Br₂ as a cocatalyst was the more effective option to suppress chain termination, although other systems showed more promising results[62]. It could be considered as an alternative, although compromises need to be made regarding the control of the polymerization process. The use of Fe-based cocatalysts would allow a wider application in technologies, as Cu-based cocatalysts require processing in reactors with less tendency.

Runs 1, 4, 5, and 6 were considered to check out the lowest amount of deactivator needed to operate in the photo-ATRP setup before it switches to conventional radical photopolymerization, caused by the domination of dead chain formation. Runs 14-17, which correspond to a Cu(II) loading between 52-6.5 ppm, show an increase in dispersity with decreasing deactivator concentration. Nevertheless, all polymers obtained successfully showed chain extension, although the amount of non-extendable chains increases with decreasing deactivator loading *vide infra*.

In general, such materials can be considered as a composition of polymers comprising chain extendable polymer chains comprising bromine and those with dead chains formed by termination. This depends on the loading with the deactivator. For example, an additional increase of **AL** slightly decreased $Đ$, which exhibited the lowest value in run 18 of all experiments shown in Table 1. An additional increase of **AL** (runs 19 and 20) did not significantly impact polymer data compared to run 18. Isolation of **AL** after polymerization indicated nearly no bound Cu(II) on the surface of **AL** while complementary experiments with FeBr₃ did (Fig. S9 in Supplementary Information). Complexation with Me₆TREN results in a much better complex stabilization, explaining the lower deposition in the **AL** surface.

These findings additionally motivated to check whether CuCl₂ also operated in this setup. Runs 21-24, representing loadings of CuCl₂ between 31 and 2 ppm, approved this showing a clear increase of $Đ$ and, therefore, also decreasing uniformity with decreasing CuCl₂ loading. This appears to be more significant with CuBr₂ as additionally shown by the block copolymerization experiments *vide infra*. A 2 ppm loading of CuCl₂ (Run 24) resulted in PMMA with a yield of 49%, $M_n$ of 54.2 kDa, and dispersity of 2.45. However, this material also contains chain extendable polymers as demonstrated by the increase of $M_n$ by 7 kDa with no significant change of

**Table 1 | AL for photoinduced CuX$_2$-catalyzed ATRP of MMA**

| Run | AL (mg/mL) | [M]:[I]:[CuX$_2$]:[L] | X | [L] | [M] | x (%) | $M_n$ (kDa) | Đ | $I_{eff}$ |
|---|---|---|---|---|---|---|---|---|---|
| 1 | 1 | 300:1:0.03:0.135 | Br⁻ | Me$_6$TREN | MMA | 36 | 35.9 | 1.31 | 0.31 |
| 2 | 1 | 300:1.5:0.03:0.135 | Br⁻ | Me$_6$TREN | MMA | 21 | 26.2 | 1.29 | 0.17 |
| 3 | 1 | 300:2:0.03:0.135 | Br⁻ | Me$_6$TREN | MMA | 13 | 26.4 | 1.26 | 0.09 |
| 4 | 1.5 | 300:1:0.03:0.135 | Br⁻ | Me$_6$TREN | MMA | 76 | 57.0 | 1.50 | 0.40 |
| 5 | 0.5 | 300:1:0.03:0.135 | Br⁻ | Me$_6$TREN | MMA | 15 | 28.0 | 1.27 | 0.17 |
| 6 | 1 | 300:1:0.015:0.0675 | Br⁻ | Me$_6$TREN | MMA | 75 | 56.2 | 1.63 | 0.41 |
| 7 | 1 | 300:1:0.06:0.27 | Br⁻ | Me$_6$TREN | MMA | 81 | 55.0 | 1.36 | 0.45 |
| 8 | 1 | 300:2:0.045:0.2025 | Br⁻ | Me$_6$TREN | MMA | 62 | 24.7 | 1.22 | 0.39 |
| 9 | 1 | 300:1:0.03:0.135 | Br⁻ | Me$_6$TREN | Styrene | 4.0 | 5.6 | 1.10 | 0.27 |
| 10 | 1 | 300:1:0.03:0.135 | Br⁻ | PMDETA | MMA | 3.6 | 41.8 | 1.46 | 0.03 |
| 11 | 1.5 | 300:1:0:0 | Br⁻ | none | MMA | 80 | 81.8 | 4.31 | |
| 12 | 1.5 | 300:2:0:0 | Br⁻ | none | MMA | 73 | 97.1 | 5.72 | |
| 13 | 0 | 300:2:0.03:0.135 | Br⁻ | Me$_6$TREN | MMA | No polymer | | | |
| 14 | 1.5 | 300:2:0.03:0.135 | Br⁻ | Me$_6$TREN | MMA | 41.2 | 20.3 | 1.25 | 0.32 |
| 15 | 1.5 | 300:2:0.015:0.0675 | Br⁻ | Me$_6$TREN | MMA | 25 | 23.0 | 1.40 | 0.19 |
| 16 | 1.5 | 300:2:0.0075:0.03375 | Br⁻ | Me$_6$TREN | MMA | 36 | 24.9 | 1.55 | 0.28 |
| 17 | 1.5 | 300:2:0.00575:0.01688 | Br⁻ | Me$_6$TREN | MMA | 52 | 29.4 | 2.25 | 0.40 |
| 18 | 2.0 | 300:2:0.03:0.135 | Br⁻ | Me$_6$TREN | MMA | 14 | 17.5 | 1.18 | 0.13 |
| 19 | 2.5 | 300:2:0.03:0.135 | Br⁻ | Me$_6$TREN | MMA | 20 | 15.9 | 1.24 | 0.20 |
| 20 | 3.0 | 300:2:0.03:0.135 | Br⁻ | Me$_6$TREN | MMA | 26 | 18.5 | 1.25 | 0.14 |
| 21 | 1.5 | 300:2:0.03:0.135 | Cl⁻ | Me$_6$TREN | MMA | 31 | 16.1 | 1.23 | 0.30 |
| 22 | 1.5 | 300:2:0.016:0.0697 | Cl⁻ | Me$_6$TREN | MMA | 12 | 24.4 | 1.28 | 0.09 |
| 23 | 1.5 | 300:2:0.006:0.0261 | Cl⁻ | Me$_6$TREN | MMA | 36 | 32.7 | 1.71 | 0.17 |
| 24 | 1.5 | 300:2:0.002:0.0087 | Cl⁻ | Me$_6$TREN | MMA | 49 | 54.2 | 2.45 | 0.14 |
| 25 | 1.5 | 300:2:0.03:0.135 | Cl⁻ | Me$_6$TREN | Styrene | 8.0 | 4.0 | 1.29 | 0.36 |

Experiments chose different experimental conditions as shown for the number average molecular weight ($M_n$) and dispersity (Đ). Reactions were conducted in 75 vol % DMSO for 24 h irradiated under blue light LED ($\lambda$ = 420 nm, 35 mW/cm$^2$). Conversion was determined gravimetrically. $M_n$ and dispersity Đ were obtained by gel permeation chromatography (GPC) in (Tetrahydrofuran) THF based on poly(methyl methacrylate) (PMMA) standards. $I_{eff}$ = $M_{n,th}$/$M_n$, $M_{n,th}$ = [M]$_0$/[I]$_0$×conversion×$M_m$+$M_{in}$ with [M]$_0$ = monomer concentration, [I]$_0$ = initiator concentration, $M_{n,th}$ = theoretical number average molecular weight, $M_m$ = molecular weight of the monomer, and $M_{in}$ = molecular weight of the initiator.

the dispersity *vide infra*. Thus, the number of polymers contributing to chain extension can be seen as small.

As expected, changing the monomer to Styrene (run 25) resulted in less conversion due to monomer's less reactivity. Nevertheless, the polymer obtained was transferred to block copolymerization experiments either, where it operated as a macroinitiator *vide infra*.

A kinetic study confirmed a linear increase in conversion during exposure (Fig. 5a). Temporal control of these experiments was performed by switching the blue light LED ON/OFF intermittently (Fig. 5b). The system showed no response in the dark, indicating the back-formation of the deactivator CuBr$_2$/Me$_6$TREN in the catalytic exposure cycles.

**Chain end fidelity**

To verify the end-group fidelity of polymers, chain extension experiments were carried out with further monomers. The macroinitiator PMMA-Br successfully extended its chain with MMA, benzyl methacrylate (BMA), and Styrene (St). Complementary experiments with different deactivator loading also approved the feasibility of chain extension, while the fraction of polymers with dead chains increased with decreasing metal halide concentration, as shown by the increase of Đ. Figure 6 illustrates the scenario using either CuBr$_2$ or CuCl$_2$ as a co-catalyst. In the case of CuBr$_2$, the second block exhibited a higher dispersity, Fig. 6a, although both blocks possessed a high average polymerization degree $P_n$. The change of the second block to BMA, Fig. 6d, gave a similar scenario considering the $P_n$ of both blocks. The use of St resulted in the expected results, where the block of poly-St exhibited a lower polymerization degree in both examples, Fig. 6e–f. The use of poly-St

as a macroinitiator, Fig. 6f, clearly demonstrates its high reactivity as shown by the larger $P_n$ of the second block; that is *b*-PMMA. This agrees with previous findings using heterogeneous photocatalysts in photo-ATRP cycles[41,42,52,53]. However, a lower loading with copper also led to a higher polymerization degree caused by the competing contribution of conventional radical polymerization.

Further experiments should focus on getting a more detailed pattern of the end group. This responsibility operates in the chain extension experiment. These can be mass spectrometric experiments, such as matrix-assisted laser desorption ionization (MALDI) mass spectrometry, which are particularly useful for halogen atoms as a narrow group, where the isotopes give a specific pattern to the molecular ion. A recent report of photo-ATRP focused on these aspects[63,64]. This gives a detailed pattern about the availability and chain-extendable groups comprising bromine and those ending in dead chains. The interpretation of MALDI mass spectrometric data also requires a critical interpretation of the peak intensities.

Furthermore, a detailed analysis of the NMR spectra with respect to the end group was not pursued because the degree of polymerization was too high to obtain a reliable number related to Mn. Figure S10 in Supplementary Information shows the NMR spectra for the two copolymers of Fig. 6d and 6e with respect to PMMA-*b*-PBMA and PMMA-*b*-PS, respectively. Based on the data obtained from the experiment, no further evaluation was carried out to avoid over-interpretation of the results. This worked better in a previous report[64]. The molecular weights reported there remained on the lower region, which enabled a more reliable data evaluation because the respective end group peaks exhibited a much larger intensity based on the signal/noise ratio.

**Fig. 5 | AL photoinduced Cu-catalyzed radical polymerization of MMA under blue light LED irradiation (λ = 420 nm, 35 mW/cm²).** Reaction condition: [MMA]/[EBPA]/[CuBr₂]/[Me₆TREN] =300/2/0.03/0.135 in DMSO 75 vol% (m_AL = 4.5 mg). **a** Kinetics of the photopolymerization, **b** Temporal control of the photoinduced ATRP process.

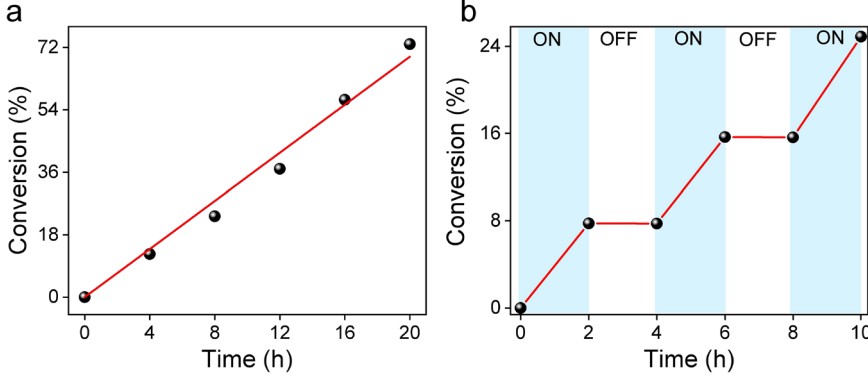

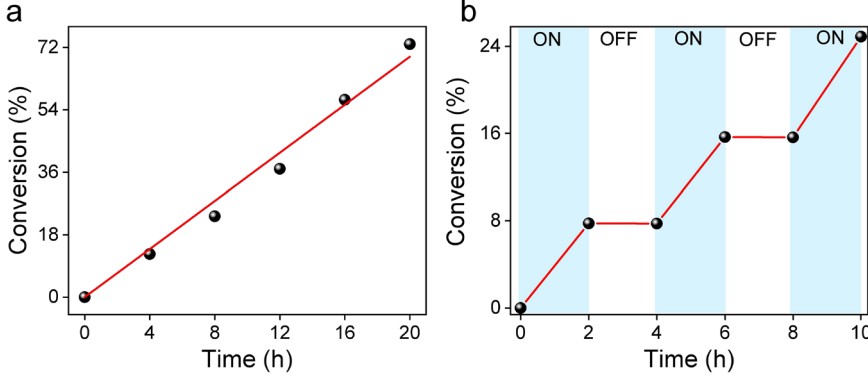

**Fig. 6 | AL photoinduced CuX₂-catalyzed radical polymerization of MMA under blue light LED irradiation (λ = 420 nm, 35 mW/cm²).** Reaction condition: [MMA]/[EBPA]/[CuX₂]/[Me₆TREN]=300/2/0.03/0.135 in DMSO 75 vol% (m_AL = 4.5 mg). GPC of PMMA and chain-extended PMMA (second block: *b*-PMMA) with different loading of CuBr₂ deactivator, **a** 52 ppm; **b** 13 ppm; **c** 6.5 ppm. Chain extension with different monomers (52 ppm CuBr₂). **d** PMMA with BMA (second block: *b*-BMA); **e** PMMA with St (second block: *b*-PS); **f** PS with MMA (second block: *b*-PMMA). Using CuCl₂ as deactivator (31 ppm loading) and different monomers results in the GPC shown in **g** PMMA and chain-extended PMMA (second block: *b*-PMMA); **h** PMMA and chain-extended PMMA with BMA (second block: *b*-BMA); **i** PS and chain-extended PS with MMA (second block: *b*-PMMA). GPC of PMMA and chain-extended PMMA (second block: *b*-PMMA) with different loading of CuCl₂ deactivator. **j** 16 ppm; **k** 6 ppm; **l** 2 ppm.

**Table 2 | Light-mediated ATRP under air condition (open air or sealed with no deaeration) sensitized by AL with 52 ppm CuBr₂**

| Run | condition | [Py]:[15-5]:[TBAB] | Conversion (%) | $M_n$ (kDa) | Đ |
|-----|-----------|--------------------|----------------|-------------|---|
| A | open air | 30:0:30 | 27 | 25.7 | 1.32 |
| B | open air | 30:30:0 | 0 | | |
| C | open air | 0:0:30 | 0 | | |
| D | sealed | 30:0:0 | 22 | 24.5 | 1.26 |
| E | sealed | 30:30:0 | 68 | 40.2 | 2.09 |
| F | sealed | 0:30:0 | 0 | | |
| G | sealed | 30:0:30 | 0 | | |

Reaction conditions: $\lambda_{exc}$ = 420 nm, $I$ = 35 mW/cm², [MMA]:[EBPA]:[CuBr₂]:[Me₆TREN] =300:2:0.03:0.135. Conversion was determined gravimetrically. $M_n$ and dispersity Đ were obtained by gel permeation chromatography (GPC) in THF based on poly(methyl methacrylate) (PMMA) standards. Exposure time: 24 h. Py: sodium pyruvate. 15-5: 15-crown-5.

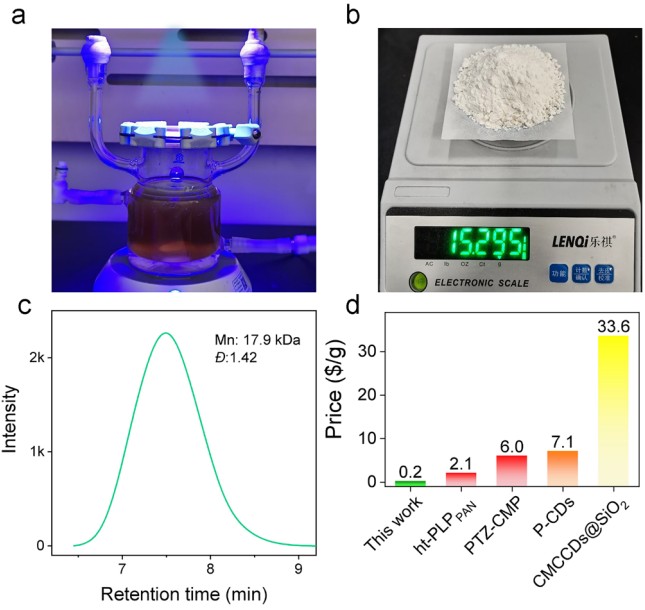

**Fig. 7 | Scale-up photo-ATRP by lignin. a** Diagram of an experimental device for scale-up photo-ATRP; **b** Photograph of PMMA produced from scale-up photo-ATRP; **c** GPC of PMMA from scale-up photo-ATRP; **d** Cost comparison of lignin with other photosensitizers; (ht-PLP_PAN = polyacrylonitrile-derived photoactive polymers[71], PTZ-CMP: Conjugated cross-linked phenothiazines[59], P-CDs: organic network of carbon dots crosslinked with porphyrin[53], CMCCDs@SiO₂ = carboxymethyl cellulose-based CDs confined in SiO₂ matrix[52]). Prices of other photosensitizers just correspond to the cost of raw materials available from the fine chemical supplier Aladdin. The yield for **CDs** was calculated as 100%.

## photo-ATRP under air

Experiments were carried out under air. The concentrations of the ATRP cocktail were taken from Run 14 in Table 1 and sodium pyruvate (Py), CH₃-COO⁻ Na⁺, was added to deactivate inhibiting oxygen in the system (required filtering of precipitated NaBr of the reaction cocktail prior to use) following a previous procedure of a non-sensitized CuBr₂/L-based photo-ATRP[65,66]. Tetrabutylammonium bromide (TBAB) addition increased bromide concentration[65,66]. Alternatively, addition of 15-crown-5 (15-5) should complex Na⁺ to prevent filtration. Table 2 shows the results obtained.

The polymer obtained exhibited a dispersity of 1.32 (Run A). Thus, ion exchange in CuBr₂/L by Py, disclosed earlier[65] followed by filtering was effective. Complexation of Na⁺ by 15-5 did not bring the expected reactivity (Run B). Run C demonstrated failure after adding of TBAB. Experiments

pursued as sealed samples with no inertization exhibited similar findings as documented by Run D. Run E resulted in a polymer with much higher dispersity and conversion. A higher contribution of conventional radical polymerization explains these findings. Obviously, 15-5 interfered with the ATRP cycle, Run F.

A mechanistic discussion should be based on the conversion of inhibiting oxygen into those with less inhibiting properties, such as $O_2^{-\bullet}$ following a previous proposal to pursue oxygen-tolerant metal-free ATRP[67]. As shown *vide supra*, the metal-free ATRP did not work with **AL**. It succeeded better even in photo-RAFT systems[29,68]. Thus, CuBr₂/Me₆TREN was added as cocatalyst. Processing under air requires extension of this proposal. The addition of Py enabled the conversion of inhibiting oxygen into an oxygen species with less inhibiting properties in the reaction cocktail. This is related to a Fenton cycle[69] where the activator formed and oxygen resulted in the formation of deactivator, and the less-inhibiting superoxide anion $O_2^{-\bullet}$ and $H_2O_2$[65,66,70]. This mechanism with Py proposed the formation of $CO_2$. Thus switched the aerobic surroundings to an anaerobic one. The back formation of the activator requires an additional electron transfer step. These discussions are based on a non-sensitized photo-ATRP system with CuBr₂/**L**. The experimental conditions of this report used lignin as a sensitizer. Here, photoexcited lignin (**AL**\*) and/or oxidized lignin, the polaron formed **AL**⁺•, may additionally react with Py, leading to the additional formation of $CO_2$ by the reaction between both. This appears much more complex, requiring complementary investigations to confirm the reaction proposed, which can be time-resolved spectroscopy, optical absorption, or paramagnetic detection of species formed. Such studies would give more insight since the oxygen scrubbing was studied with a non-sensitized system. Although such sensitized systems appear very practical, there would be more fundamental research required to bring them to practical applications. Furthermore, experiments pursued with CuCl₂ instead of CuBr₂ showed no polymer formation under aerobic conditions.

### Practicability - scale-up of photo-ATRP

Figure 7a shows the scale-up experimental device. A scale-up to 100 mL resulted in 15.3 g of PMMA, corresponding to a gravimetric yield of 65% see Fig. 7b. GPC resulted in moderately lower $\bar{M}_n$ but slightly higher Đ compared with the experimental condition in run 14 (Fig. 7c). The photo-ATRP depicted here used a batch reactor, whose size was adjusted according to the scale-up factor of 25. Larger thickness in the reactor changes absorption conditions and, therefore, polymer radical concentration. This may cause slightly increased termination. Nevertheless, it confirmed that the photo-ATRP initiated by lignin can be scaled up, which provides a promising potential for photo-ATRP sensitized by biomass. More importantly, as a renewable biomass resource, lignin is not only non-toxic but also possesses great advantages in terms of cost. Figure 7d exhibits a comparison of costs for lignin as a heterogeneous photosensitizer, which appeared as at least one-tenth compared to that of other photosensitizers.

## Conclusions

**AL**, available on a large scale as waste in the paper-making industry, possesses, according to its functional groups, electrochemical and photonic properties enabling it for radical photopolymerization based on conventional and controlled radical polymerization protocols. The latter employed a photo-ATRP protocol based on CuBr₂/**L** operating as a cocatalyst with blue light LED exposure, with the feature of block copolymerization. These polymerizations only require an amount of Cu(II) cocatalyst between 5-10 ppm demonstrating the potential for future work. Such loadings cannot be seen anymore as harmful for many uses of the polymers made. The transfer of the reaction protocol successfully worked at a larger scale, demonstrating the practicability of the reaction system.

**AL** as heterogeneous sensitizer enables radical photopolymerization in air surroundings based on the photo-ATRP protocol in the presence of pyruvate as an oxygen-scrubbing agent. This appears challenging because controlled polymerization conditions must compete with reactive oxygen species (ROS), which can easily interfere with the catalytic ATRP cycle.

Thus, the setup of optimal oxygen scrubbing still appears challenging. Interestingly, the heterogeneous matter of **AL** additionally questions whether such systems possess, in general, more robustness compared to homogeneous systems where a single molecule operates the process. More research would be required to explore this mechanism, although the first reports provide the direction to set the focus. Thus, it would be desirable to get answers about the formation of $CO_2$, which can contribute to form an anaerobic reaction environment. Here, the $CO_2^{-\bullet}$ radical, one of the strongest reduction agents in organic chemistry, possesses a key function to run radical polymerization under air.

Additional work in the future should focus on providing materials that can be used with improved efficiency to initiate conventional radical photopolymerization operating as a one-component photoinitiator. Functionalization with hydroxyl moieties with additional photocleavable groups would be one alternative, although a decrease of hydrophilicity by incorporating an alternative would also give promising future directions. In addition, selective oxidation of the benzyl carbon, resulting in carbonyl groups, would enhance the number of photocleavable groups. These intermediates formed possess a better diffusion ability compared to photoproducts formed by non-modified lignin. Thus, the functionalization provided the direction where future work could set more research activities.

This study demonstrates the successful use of biomass as a photosensitizer for photopolymerization. Lignin does not exhibit any toxicological issues, but its availability as waste in this study requires screening of reactivity priority. This depends generation of Lignin in the production, where it is available, and may depend on the industrial location. Nevertheless, it is an interesting source to operate as a photosensitizer. Future studies will focus on exploring the relation between the surface topology of heterogeneous photocatalysts and their reactivity and photocatalytic polymerization protocols using $CuX_2$/**L** as a cocatalyst. This point requires more attention in future studies, particularly the understanding of adsorption/desorption processes proceeding on the surface of the photocatalyst. This photocatalyst appears heterogeneous and addresses challenges regarding the understanding of chemical reactions. To understand the polymerization process, adsorption and desorption must be considered. Exploring the surface properties of photocatalysts in greater detail will provide more insights into the reaction conditions.

In general, lignin is an interesting class of substances that is suitable for initiating photoreactions. This can be done as a waste product, as demonstrated in this work. Further modifications and functionalization of classes of interesting developments are expected in the future to use it as a basis for the development of photoinitiating materials for conventional and controlled radical polymerization.

## Methods
### Materials
**General information.** **AL**, methyl methacrylate (MMA), benzyl methacrylate (BMA), and styrene (St) Hydroxyethyl acrylate (HEA, 96%) and Urethane dimethacrylate, a mixture of isomers (UDMA, >97%) were purchased from Aladdin. Dimethyl sulfoxide (DMSO, anhydrous, 99.8%), Copper(II) bromide(CuBr$_2$), Iron (III) bromide (FeBr$_3$), ethyl $\alpha$-bromophenylacetate (EBPA, 97%), Tris[2-(dimethylamino)ethyl] amine (Me$_6$TREN, >98.0%), $N,N,N',N'',N''$-pentamethyldiethylenetriamine (PMDETA, 99%), Tetrabutylammonium bromide (TBABr, 99.0%), Benzenesulfonyl chloride (98%), and triethylamine (TEA) were obtained from Aladdin. Dichloromethane (DCM), tetrahydrofuran (THF), and methanol (99.9%) were purchased from Aladdin. Monomers except UDMA, were passed through a column filled with neutral aluminum oxide to remove the inhibitor before use. All other chemicals needed for the experiments were received from Aladdin as well and used as received.

**Synthesis of arylsulfonated lignin (AL-SO$_3$).** The synthesis of arylsulfonated lignin was carried out following a disclosed procedure[1]. Specifically, 1 g of **AL** was put in a reaction bottle and first degassed by vacuum and then filled with nitrogen. Next, 20 mL dichloromethane (DCM) was injected in the bottle to disperse the **AL** followed by addition of 1.4 mL (10 mmol) of TEA followed by stirring for about 20 min. After that, the reaction bottle was cooled in an ice bath before 1.3 mL (10 mmol) benzene sulfonyl chloride in 10 mL DCM was slowly added drop by drop using an injection syringe. Finally, the reaction mixture was kept at room temperature for 4 h. At the end of the reaction, the solid product was separated and washed with deionized water three times, followed by drying before use. Figure S11 in Supplementary Information shows successful aryl sulfonation of Lignin resulting in an increase in the sulfur content.

### Characterization methods
**UV-Visible spectroscopy.** UV-Vis spectra **AL** dispersed in DMSO were recorded by a PERSEE TU-1950 UV-Vis Spectrophotometer in $1 \times 1$ cm cuvettes.

**Transmission Electron Microscopy (TEM).** TEM was conducted using a transmission electron microscope (Talos F200S G2). **AL** was dispersed in DMSO to obtain the concentration of 10 ppm by ultrasonic treatment.

**Infrared Spectroscopy (IR).** IR spectra were recorded using a Fourier transform infrared spectrometer (Thermo Fisher Scientific) with a resolution ratio of 4 cm$^{-1}$ and a scan range of 4000~400 cm$^{-1}$ using the attenuated total reflection (ATR) method. The solid powder sample was directly put on the test board and then tested.

**$^1$H-NMR Spectroscopy.** Fourier 300 from Bruker was used for all $^1$H-NMR measurements. 10 mg sample were placed in 0.7 mL deuterated chloroform (CDCl$_3$).

**X-ray Photoelectron Spectroscopy (XPS).** X-ray photoelectron spectroscopy (XPS) was performed using a thermal spectrometer (Thermo Fisher Scientific, ESCALAB 250Xi) with a monochromatic Al Kα X-ray source. Ceshigo research service pursued the experiments.

**Cyclic voltammetry (CV).** Electrochemical experiments were performed with a CHI660E potentiate (Chenhua Company, Shanghai, China) in a conventional three-electrode cell. The electrode assembly consists of a platinum wire as the counter electrode, a standard calomel electrode(SCE) as the reference electrode, and a glass carbon (GC) electrode as the working electrode. The measurement buffer was 0.5 M Na$_2$SO$_4$. Cyclic voltammetry (CV) was recorded in the range from $-1.6$ V to 2 V with a scan rate of 0.05 V s$^{-1}$. To compare the samples, 1 mg **AL** was dissolved in 1 mL DMSO with CuBr$_2$ solution (9.4 µL, 100 mM, DMSO) and Me$_6$TREN (1.1 µL) and finally dispersed evenly by ultrasonic, then dropped 100 mL of sample solution evenly on the conductive glass and dried at 80 °C to get the sample for testing.

**Photocurrent measurement.** The photocurrent was measured using CHI660E potentiostat (Chenhua Company, Shanghai, China). **AL** solution (800 µL, 5 mg mL$^{-1}$, H$_2$O/C$_2$H$_5$OH (1:1, v/v)) was mixed with naphthol (100 µL) assisted by sonification for 30 min. After that, the 100 µL mixture was dropped on ITO-coated (ITO: Indium Tin Oxide) glass and dried under air for 24 h. The coated ITO glass worked as the working electrode and Na$_2$SO$_4$ (0.5 M) was used as bthe uffer. The photocurrent was recorded under Xenon lamp irradiation (100 mW/cm$^2$).

**Gel Permeation Chromatography (GPC).** Gel Permeation Chromatography (GPC) measurement was performed at GPC (Agilent 1100 Series) and two columns of KF-802 and KF-804 (8 mm ID*300mmL), a column temperature of 30 °C, a RI detector, and tetrahydrofuran (THF) as an eluent at a flow rate of 1 mL min$^{-1}$. PMMA standards were used for calibration.

**Article**

## Experimental Procedures

**General description of the photo ATRP setup**. The photo-ATRP procedure was conducted with the photocatalytic parallel reactor (CEL-LAB200E7, CEAULIGHT, Beijing, China). Typically, a Schlenk flask with a magnetic stirrer was degassed by four vacuum pump-nitrogen cycles. It followed the addition of **AL** dispersed in 3 mL DMSO ($c = 1.5$ mg mL$^{-1}$), which was mixed with MMA (1 mL), CuBr$_2$ solution (9.4 μL, $c = 100$ mM, DMSO), Me$_6$TREN (1.1 μL) and EBPA (11 μL). The mixture was irradiated during stirring at 420 nm (intensity: 35 mW/cm$^2$) for 24 h. After exposure, the polymers were precipitated in cold methanol, and isolated by filtration. After drying, the isolated material was dissolved again in THF during stirring. The solution was put into cold methanol again to precipitate the polymers followed by repeating the drying step above. The collected polymers were dried in the vacuum oven at 50 °C for 24 h. Conversion was gravimetrically determined.

**Chain Extension Procedure for photo-ATRP**. A Schlenk flask with a magnetic stirrer was degassed with a vacuum pump followed by nitrogen exchange in repeated four cycles. Afterward, **AL** solution (1.5 mg mL$^{-1}$, 3 mL, DMSO) was mixed with MMA (1 mL), CuBr$_2$ solution (9.4 μL, 100 mM, DMSO), Me$_6$TREN (1.1 μL) and macroinitiator PMMA-Br (1200 mg). The mixture was irradiated under stirring at 420 nm for 24 h. After exposure, the polymers were precipitated in cold methanol followed by isolation of the polymer. After drying, the isolated polymer was dissolved again in THF. The solution was put into cold methanol again to precipitate the polymers. The collected polymers were dried in the vacuum oven at 50 °C for 24 h.

**Block Copolymerization Procedure for photo-ATRP**. The block copolymerization procedure was mostly identical as disclosed above but some minor changes were made. Firstly, the Schlenk flask equipped with a magnetic stirrer was degassed by four vacuum pump-nitrogen purging cycles. Then, a solution comprising **AL** (1.5 mg mL$^{-1}$, 3 mL, DMSO) was mixed with BMA (1.5 mL), CuBr$_2$ solution (9.4 μL, 100 mM, DMSO), Me$_6$TREN (1.1 μL) and macroinitiator PMMA-Br (1200 mg) followed by degassing. The mixture was irradiated at 420 nm for 24 h during stirring. After exposure, the polymers were precipitated in cold methanol. After drying the isolated material under air, it was dissolved in THF while stirring again. The solution was put into cold methanol again to precipitate the polymers. The collected polymer (PMMA-*b*-PBMA) was dried in the vacuum oven at 50 °C for 24 h. Another block copolymerization experiment used PMMA-Br as the macroinitiator and styrene (1.1 mL) as the second monomer, to obtain the polymers PMMA-*b*-PS. The operation procedure was the same as above. Accordingly, using PS-Br as the macroinitiator and MMA (1 mL) as the second monomer, the polymer PS-*b*-PMMA was obtained.

**ON-OFF Procedure in photo-ATRP**. The ON-OFF experiments were conducted in five separate reactor sites with photocatalytic parallel reactors. As described above, after degassing and adding the starting material, all five Schlenk flasks were put into the reactor. Exposure was carried out for two hours with 420 nm irradiation, followed by the removal of the reaction tube to collect the product. After two hours of dark treatment, the reaction tube was brought back, and exposure started again for two hours. This procedure was repeated for all polymer samples. Polymers were collected as mentioned abov,e and conversion was gravimetrically determined.

**Real-time FTIR to determine double conversion in conventional radical photopolymerization**. Real-time FTIR (Vertex 70 from Bruker) in an attenuated total reflection (ATR) mode was applied to investigate the polymerization kinetics during light exposure (365 nm LED: 200 mW/cm$^2$) using the synthesized AL-SO$_3$ as one-component initiator. The respective mixture of monomers was first prepared. Then AL-SO$_3$ with respective amount was added to the monomer mixture followed by

treatment in an ultra-sonic bath for 30 mins. For these measurements. The obtained dispersions were put on the FTIR sample plate with a spacer of 40 μm and covered by a micro glass plate to get rid of the airflow during the radiation. In this procedure, every single data point was calculated automatically by the software (OPUS) based on the FTIR spectra from 6 scans, which were averaged and saved as a single data point. Furthermore, the data points for subsequent analysis were continuously collected every 0.36 seconds in this work. In addition, the peak at around 810 cm$^{-1}$ indicating $-C=C-$ was taken for the evaluation of reaction kinetics and final conversions while the peak at 1700 cm$^{-1}$ representing $-C=O$ as reference. This method was evaluated over the years and it was found very reproducible. Typical errors responsibly leading to differences can be seen in the balance ($+/- \ 0.005$ mg), spacer adjustment ($+/-2$ μm), ultra-sonic treatment to achieve satisfied dispersion of material, and the light intensity ($+/- \ 5\%$). The latter was taken with a fiber optical spectrometer USB 4000 from Ocean Optics.

An experiment with **AL** failed due to insufficient dispersibility.

**Screening radical polymerization by arylsulfonated lignin**. 1 mg of arylsulfonated lignin (AL-SO$_3$) was added to 300 mg HEA and dissolved completely by ultrasonic treatment. Then, 700 mg UDMA was put into this solution resulting in a uniform mixture. Then, the mixture was put on glass sheets wrapped with 6 layers of transparent tape on the edge, and then covered by another glass sheet on the mixture. This was fixed with clips. Finally, these glass sheets were treated with 365 nm UV light exposure for 20 min to obtain a photopolymerized material. The successful experiment followed a kinetic study.

## Data availability

The authors declare that the data supporting the findings of this study are available within the paper and its supplementary information files. Source data for Figs. 2–7 are provided in Supplementary Data 1. All data underlying this study are available from the corresponding author Bernd Strehmel upon request.

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

## Acknowledgements
ZC acknowledges the National Natural Science Foundation of China (32171716) for supporting his research. He also acknowledges the funding of Central University (2572022CG02). XL is grateful for the funding of the National Natural Science Foundation of China (32301534) and Central University (2572023CT11). SL acknowledge the National Natural Science Foundation of China for financial support associated with grant numbers 32271795. BS and VS acknowledge Niederrhein University of Applied Sciences for financial support of research.

## Author contributions
Conceptualization: X.L., Z.C., S.L., B.S.; Methodology: B.S., X.L., V.S.; Investigation: M.W., Q.W.; Visualization: M.W., X.L., Q.W.; Supervision: S.L., Z.C., B.S.; Writing-original draft: All authors; Writing-review and editing: All authors.

## Funding

## Competing interests
The authors declare no competing interests.
