## [Transparent Peer Review file · Communications Chemistry]

This manuscript has been previously reviewed at another journal. This document only contains information relating to versions considered at Communications Chemistry.

Valorization of Soda Lignin: Sustainable Photosensitive Component for Conventional and Controlled Radical Photopolymerization

Corresponding Author: Professor Bernd Strehmel

Version 0:

Reviewer comments:

Reviewer #1

(Remarks to the Author)

This publication describes the use of a modified Lignine (AL) as a photosensitizer for an ATRP chemistry. Although ATRP is well known and has been exhaustively used and described, the use of a waste product as a photoinitiating (PI) system is attractive and merits attention.

This is a well conducted paper, with many interesting details considered. Overall, also after reading the comments of the first review round, I feel that this paper should be published in this very journal, as the results are novel, the concept attractive and technically useful, thirdly representing an innovation in the field, despite the known photo-ATRP chemistries well published.

I comment on the following:

- 1) The endgroup concept (see line 279 and thereafter) is not fully conclusive (supported by data) - endgroup analysis requires a deep MS-analysis, together with an exhaustive NMR analysis of the formed polymers. I did not see this in this manuscript profoundly, and recommend (a) either to reduce the intensity of this part (wording here is important), or provide the required MS/MALDI-TOF and NMR evidence for it. My suggestion would be - as it is so well placed as an own part in the main manuscript - to place original data for this in the main part as a Figure.
- 2) Lignine and the resulting degraded lignines are always (chemically) heterogeneous - depending on (a) the source and (b) the method of degradation and (c) the experimentalist conducting the degradation. This seems not to be addressed at all in this manuscript, and I strongly recommend to address it. I would presume that the number of eg. phenolic units/CH₂OH units is relevant for the subsequent photo-ATRP. Overall also reproducibility would be considered, and this would be an important part for this manuscript.
- 3) On some occasions the manuscript reads very dense - see eg. the electrochemical part. I appreciate this, but just recommend to ease these parts, so that readability is improved for the readers.

Apart from this I think this is an excellent manuscript, which - after appropriate improvements as suggested above - can be published in this journal.

Version 1:

Reviewer comments:

Reviewer #1

(Remarks to the Author)

The authors have revised the manuscript, and taken their point(s), which I can accept, also in view of the exhaustive work conducted here. Although I do not feel that MALDI TOF is a specific technique in polymer science any more (its standard, to be honest), the authors have modified their statements such that it seems acceptable.

In the future though, when heading a chapter "endgroup fidelity" MALDI TOF analysis must be provided, because otherwise the general notion of a polymer based audience would be misled.

The paper can be accepted for publication.

Thank you for reviewing this manuscript.

1) The endgroup concept (see line 279 and thereafter) is not fully conclusive (supported by data) - endgroup analysis requires a deep MS-analysis, together with an exhaustive NMR analysis of the formed polymers. I did not see this in this manuscript profoundly, and recommend (a) either to reduce the intensity of this part (wording here is important), or provide the required MS/MALDI-TOF and NMR evidence for it. My suggestion would be - as it is so well placed as an own part in the main manuscript - to place original data for this in the main part as a Figure.

That is a very valuable comment. Indeed, with GPC, the molar masses are related to PMMA and a more exact evaluation of contributing termination reactions appears as critical. MALDI is of course a very effective method to get direct access to molar masses, but one should always keep in mind that an evaluation of the distribution to obtain the dispersity involves much more additional experimental work.

We have cited relevant work in the manuscript and indicated that future work will include such experiments. At the time of submission of our manuscript, our new MALDI was not yet functionally ready. Therefore, such experiments could not be performed. However, this is also a method that is not available to everyone. It has become standard practice to determine molar masses with GPC, even if this has to be calibrated against a standard. Nevertheless, we have incorporated these points into the manuscript at the relevant point.

An analysis of the in-group NMR spectra showed that the corresponding signals were too small to extract reliable data or information. The molar masses of the polymers were significantly higher compared to the corresponding end groups. Of course, this is a very effective method. However, it was used for polymers in the cited reference that had smaller molecular weights. We think that it is therefore not appropriate in this study, although it was well meant by reviewer, to include these considerations. We have also incorporated these critical remarks into the manuscript and expanded the supporting information accordingly with the NMR spectra. We also uploaded the respective original NMR data if someone would have a deeper interest to view them.

2) Lignine and the resulting degraded lignines are always (chemically) heterogeneous - depending on

(a) the source and (b) the method of degradation and

This is very valuable information, which we have incorporated at the appropriate points in the manuscript. We have also pointed out that different photocatalytic reactivities may be expected depending on batch to batch variation. This is a general problem of these sustainable materials which has not received the necessary attention in the community yet.

(c) the experimentalist conducting the degradation. This seems not to be addressed at all in this manuscript, and I strongly recommend to address it. I would presume that the number of eg. phenolic units/CH₂OH units is relevant for the subsequent photo-ATRP.

We think the scenario is much more complicated. Even these groups form supramolecular structures which mainly are responsible for the studied photocatalytic activity. This was shown in our studies of photopolymerization using another matrix, which was cited. Although this was good meant by the reviewer, we prefer to briefly discuss the scenario

because we do not have enough reliable data for deeper disclosure. Of course, the fact radical polymerization worked seems to be a great result, because phenolic groups and C-radicals are typically like "cat and dog".

Overall also reproducibility would be considered, and this would be an important part for this manuscript.

The experiments were reproducible with acceptable results although these were heterogeneous systems where many parameters affect the photoinduced process such as scattering and others. For Run 14, the following data were obtained:

Run	AL (mg/mL)	[M]:[I]:[CuX ₂]:[L]	X ⁻	[L]	[M]	x (%)	M _n (kDa)	D
1 (Run 14)	1.5	300:2:0.03:0.135	Br ⁻	Me ₆ TREN	MMA	41.2	20.3	1.25
2	1.5	300:2:0.03:0.135	Br ⁻	Me ₆ TREN	MMA	46.0	20.4	1.24

The system seems to be robust because synthesis at large scale also worked and brought the expected outcome on results.

We added at appropriate places in the manuscript these points and noticed the complexity of heterogeneous systems compared to those operating in a homogeneous system. These heterogeneous systems are still in their infancy.

3) On some occasions the manuscript reads very dense - see eg. the electrochemical part. I appreciate this, but just recommend to ease these parts, so that readability is improved for the readers.

We did this in the electrochemical part and some other parts in the manuscript.

We also read the manuscript again and found it readable and understandable for the readership of Communications Chemistry.

Thanks again to this reviewer who spent a large amount of his time to improve the quality of this manuscript. We took these comments seriously and integrated them accordingly in the manuscript wherever it fit.